# Integration of Horizontal and Vertical Microfluidic Modules for Core-Shell Droplet Generation and Chemical Application

**DOI:** 10.3390/mi10090613

**Published:** 2019-09-15

**Authors:** Dong Hyun Yoon, Yoshito Nozaki, Daiki Tanaka, Tetsushi Sekiguchi, Shuichi Shoji

**Affiliations:** 1Research Organization for Nano & Life Innovation, Waseda University, 513, Tsurumaki-cho, Waseda, Shinjuku-ku, Tokyo 162-0041, Japan; y.nozaki@aoni.waseda.jp (Y.N.); d.tanaka@ruri.waseda.jp (D.T.); t-sekiguchi@waseda.jp (T.S.); 2Faculty of Science and Engineering, Waseda University, 3-4-1, Okubo, Shinjuku-ku, Tokyo 169-8555, Japan; shojis@waseda.jp

**Keywords:** three-dimensional structure, device integration, microdroplet, double emulsion

## Abstract

This paper presents a method for utilizing three-dimensional microfluidic channels fully to realize multiple functions in a single device. The final device structure was achieved by combining three independent modules that consisted of horizontal and vertical channels. The device allowed for the one-step generation of water-in-oil-in-water droplets without the need for partial treatment of the polydimethylsiloxane channel surface using separate modules for generating water-in-oil droplets on the horizontal plane and oil-in-water droplets on the vertical plane. The second vertically structured module provided an efficient flow for the generation of highly wettable liquid droplets, and tuning of the first horizontally structured module enabled different modes of inner-core encapsulation within the oil shell. The successful integration of the vertical and horizontal channels for core-shell droplet generation and the chemical synthesis of a metal complex within the droplets were evaluated. The proposed approach of integrating independent modules will expand and enhance the functions of microfluidic platforms.

## 1. Introduction

Chemical reactions or synthesis are critically influenced by environmental factors, where the mixing method or atmospheric conditions can affect the reaction’s speed results and efficiency [1,2,3]. Therefore, numerous systems have been investigated and developed to improve upon conventional chemical reactions, as well as to realize novel synthesis methods [4,5].

Droplet-based microfluidic research has also developed with the fields of chemistry and biology [6,7,8,9,10]. Microdroplets allow for the fast mixing of reagents on the small scale and the individualization of specific targets using immiscible fluids. Furthermore, the small volume and high frequency of generation of the droplets have allowed for numerous experiments to be carried out in miniaturized platforms without much energy and reagent consumption

Along with the diversification of droplet research, the required type, materials, and method of manipulation of the droplets are also complicated. For instance, multiphase/layered droplets with composite materials allow for the synthesis of functional materials or the complex manipulation of target samples [11]. Water-in-oil-in-water (W/O/W) or oil-in-water-in-oil (O/W/O) droplet generation is one of the advanced methods related to droplet technology [12,13,14,15,16,17]. The shell layer efficiently isolates the inner core, and the layer can be used functionally. Furthermore, users can have a choice of the outer phase, which is responsible for the most volume and is in direct contact with the environment, for suitable operation.

Multiphase core-shell droplets have been generated by various methods. A needle inserted into a tube [18,19], a glass capillary-based device [20,21,22,23,24,25], and microchannels [26,27,28,29,30,31,32,33,34] are the representative methods and device materials. The needle insertion and glass capillary methods form the channel structure through combination of the pipe-like structures, whereas the microchannel method utilizes a micromachining process. Although the two methods of combining the pipe-like structures allow simple and rapid manufacturing of the channels, the methods are limited in the channel geometry and precision. In contrast, the microchannel method is advantageous for the channel precision and design, but these two aspects are limited to a two-dimensional structure.

Because the combination of needles in tubes or glass capillaries provides a stronger three-dimensional focusing flow, droplet generation is more convenient. However, the channel materials and structures only serve as a droplet generator and have low precision. It is fundamentally difficult to modify the channel design with these methods and to realize various fluidic structures for novel functions. Furthermore, their integration with other fluidic elements is also limited.

In contrast, the two-dimensional microchannels can be designed simply and fabricated precisely. However, the two-dimensional focusing flow is not sufficient for multiphase droplet generation. Because a strong focusing flow or modification of the wettability between the liquid and the channel can improve droplet generation, sophisticated methods using semi-three-dimensional flow [26,27] or treatment for partial modification of the channel surface [28,29,30,31,32,33,34] have been developed.

In our previous study [35], we had also developed a multiphase droplet generating method based on a functional structure. Using only structural features, fully three-dimensional focusing flow was realized, and the effective focusing flow allowed the generation of highly wetting liquid droplets on the channel surface without the need for additional surface treatment.

However, the research has mainly been focused on droplet generation. In many fields, applications using multilayer droplets require not only their generation but also other manipulations and complicated processes [36].

Hence, we have developed a method for modularizing the functional structures and integrating the modules into a single device. Specifically, the main module was designed for multiphase droplet generation, and the feasibility of pre- and post-integration of the module with other elements designed for different functions was evaluated. By using this method, an all-in-one operation for complex droplet generation and their pre- or post-manipulation is made possible. This will realize more complex and functional chemical or biological platforms, as well as novel devices for the material sciences.

The development was evaluated as follows. First, the efficiency of the droplet-generating modules was verified by using them to generate droplets of different oil types that are highly wetting on a channel surface. Through this verification, we showed the utility of a droplet generation module formed by a structure that can be integrated with other elements. Second, following integration of the fluidic modules with their individual functions, evaluation of the functional diversity and operational feasibility of the device was carried out. Finally, the applicability of this system was verified by conducting a chemical reaction in the droplets using the integrated device.

## 2. Principle and Design

Herein, we propose three types of modules for realizing three different functions, as shown in Figure 1. Modules 1, 2, and 3 are intended for water-in-oil (W/O) droplet formation, oil-in-water (O/W) droplet formation, and mixing or observation, respectively. By combining these modules, the continuous one-step generation of W/O/W droplets and analyzing of inner cores in the droplets can be achieved in a single device.

The first module comprises two cross channels connected to each other. This structure forms two types of droplets independently and brings the droplet together, thus it allows the preparation of the different reagents. The second module consists of a protruded cylindrical channel end and a three-dimensional focusing channel. The channel provides a fully three-dimensional focusing flow and the efficient focusing flow enables droplet generation of highly wetting liquid on the channel surface. This feature expands a limit of the applicable materials without complex consideration for the conditions of the liquid and channel surface. Finally, the third module has a vertically bent and horizontally curved channel for simple and sufficient observation purposes. The simple structure can be exchange with conventionally well-defined functional channels such as sensors and actuators. Therefore, integration of the three modules provides preparation of independent reagents as a droplet phase, encapsulation of the reagents in a single W/O/W droplet, and observation or additional manipulation, sequentially.

Figure 2 shows the schematic view and detailed dimensions of the modules and the fully integrated device. Two types of Module 1 were proposed to obtain different W/O droplet generation modes and for comparing their effects on the W/O/W droplet generation results. Module 2 was constructed by combining two fluidic units, where units 1 and 2 formed a protruded channel end and four branched focusing channels. The structure provides for both vertical injection and radial focusing of the core flow. The structure of Module 2 allowed an efficient formation of microdroplets even in the case of highly wetting liquid. However, we had evaluated droplet generation in the three-dimensional channel with only a straight downstream channel in our previous study [35]. Although functions of the straight structure were successfully evaluated, immediate and short channel bending is essential for integration of the vertical structure with horizontal structures. The downstream channel between the end of Module 2 and the inlet of Module 3 proposed in this research is short and angular enough to affect droplet generation. Hence, the droplet generation results at the junction area in Module 2 and the function of the entire device was evaluated locally and comprehensively.

## 3. Experimental

### 3.1. Device Fabrication

The device was fabricated through the stacking of polydimethylsiloxane (PDMS) modules (SILPOT 184, Dow Corning Toray Co., Ltd., Tokyo, Japan). All PDMS modules were formed from SU-8 molds (SU-8 3025 and 3050, MicroChem Corp., Westborough, MA, USA). Four types of SU-8 molds consisting of two layers were fabricated by the UV lithography process. To generate through-holes and alignment of the modules, PDMS sheets were used in the module molding process [37]. The PDMS modules were aligned and bonded manually after oxygen plasma treatment. The results of stacking of the modules are shown in Figure 3.

### 3.2. Materials and Experimental Set-Up

For O/W droplet generation, mineral oil (M8410, CAS 8042-47-5, Sigma-Aldrich, St. Louis, MO, USA), liquid paraffin oil (Cat. No. 32033-00, CAS 8012-95-1, Kanto Chemical Co., Inc., Tokyo, Japan), and silicone oil (KF-96-10CS and KF-96-100CS, Shin-Etsu Chemical Co., Ltd., Tokyo, Japan) were employed for the oil phase flow, and deionized (DI) water with methylene blue dye (Cat. No. 25249-30, CAS 7220-79-3, Kanto Chemical Co., Inc.) was used for the water phase fluid. Moreover, 1 wt.% Tween20 (P1379, CAS 9005-64-5, Sigma-Aldrich) was mixed into the water to act as a surfactant for preventing downstream merging of the droplets after their generation.

In the case of W/O/W droplet generation, DI water and mineral oil were mainly used. As the inner water cores, water with two differently colored food dyes (red and green, Kyoritsu-Foods Co., Ltd.) was used. Mineral oil dissolved in 1 wt.% Span 80 (Cat. No. 37408-32, CAS 1338-43-8, Kanto Chemical Co., Inc.), and DI water dissolved in 3 wt.% Tween20 were used as the liquids for the oil shell and outer carrier flow.

Finally, chemical synthesis in the W/O/W droplets was tested. For this purpose, 20 mM copper(II) acetate monohydrate (MW 199.65) and a ligand produced from 40 mM 3,5-dichlorosalicylaldehyde (MW 191.01) and 20 mM (1R,2R)-(+)-1,2-diphenylethylenediamine (MW 212.29) were used for the synthesis of copper complexes within the droplets.

All liquids were introduced into the device via syringes (1725CX or 1750CX, Hamilton, Reno, NV, USA) and syringe pumps (KDS, KD Scientific Inc., Holliston, MA, USA). The droplet generation process and flow were observed with a high-speed camera (FASTCAM-NEO, Photron, Tokyo, Japan). The droplet sizes were evaluated and calculated by pixel counting of the captured images.

## 4. Results and Discussion

### 4.1. O/W Droplet Generation

The O/W droplet generation results are shown in Figure 4. Mineral oil injected from the first module flowed into the vertical cylindrical channel of the second module, where the oil flow was focused by the water flow in the second module. Therefore, the oil droplets were formed at the junction of Modules 2 and 3 (Appendix A). 

Although the oil was highly wetted to the PDMS surface, the full-side focusing flow of the water effectively suppressed spreading of the oil on the module surface, and thus O/W droplets could form without the need for hydrophilic coating or treatment of the channel surface. Furthermore, the continuous carrier flow prevented sticking of the oil droplets downstream of the channel (Figure 4A) and allowed a larger droplet volume of mineral oil than the channel width (Figure 4C). The function of the droplet-generating structure integrated with the protruded junction end and bent channel was then evaluated. We had employed a straight downstream channel after flow focusing in our previous study [35]. In contrast, this present study is the first to show that the protrusion structure is effective in forming droplets of highly wettable liquids on channel surfaces, even in a channel where the focused flow is immediately bent.

Droplet generation was evaluated according to different flow rates and oil types (Figure 5) using mineral oil, liquid paraffin oil, and silicone oil. The proposed structure successfully allowed the generation of droplets of the various oils. Furthermore, the droplet size increased with increase in the flow rate of the sample phase liquid, whereas it decreased with increase in the flow rate of the carrier flow similar to the general droplet generation results. However, there was a limit to the fluidic conditions that droplets could be formed in all cases because of the oil’s high wettability on the channel surface. Significantly, in the case of mineral oil, the droplets could be generated under oil flow rates of slower than 1 μL/min, and the droplet size formable was larger than that of the other oils (Figure 5A). In addition, the maximum generation rate of mineral oil droplets was 70 drops/s, when the flow rates of oil and water were 0.1 and 20 μL/min, respectively.

In contrast, the other oils (Figure 5B–D) required the carrier flow rate of the water phase to be 10–20 times higher than that of the oil phase for droplet generation. It seems that the relatively low wettability of mineral oil enabled a wide range of droplet generation. Moreover, the droplet size was proportional to the viscosity of the compared oils. When the oil flow rates exceeded their limits, water droplets were generated in the device instead, but high flow rates of faster than 20 μL/min could wash the oil’s wetting in the junction area and allows oil droplet generation again.

The flow field in this device is different from that of conventional device models such as T-junction or cross-channel because the structure suddenly forms a bent flow after radial focusing flow. In addition, the working fluids can be easily wetted onto the channel surface and the geometric distance between the protrusion end of the second module and the bottom surface of the third module is very short. The structural and material features formed the complicated droplet generation results. Although the absolute volume flow rate of the injected oil was concluded to be more dominant than the carrier flow rate and the wettability of oil greatly affects the limit conditions for the formation of droplets, additional and detailed investigations of the correlation between the droplet generation results and related parameters are necessary.

### 4.2. W/O/W Droplet Generation

On the basis of the results above, W/O/W droplets were generated via two types of Module 1. Although both types A and B consisted of four injection channels, they were previously designed to generate two different types of droplets and to inject four different fluids into the single junction simultaneously.

As shown in Figure 6 (Appendix A), the droplets were generated independently in the first module and then flowed down into the protruded channel. Then, the water droplets were encapsulated in W/O/W droplets while O/W droplets were generated in the second module.

In the case of using Module 1-A, the two independently generated droplets were randomly injected into the second module. Therefore, the morphology of the W/O/W droplets changed depending on the generation and injection timing. Furthermore, some water droplets that had formed in the first module were broken again at the junction area owing to the orthogonal channel structure and additional carrier flow from the opposite channel. Consequently, the W/O/W droplets generated in the device ranged from a single core to four cores under the flow conditions.

In contrast, when the immiscible liquids were joined at a single junction and the flow was focused again, W/O/W droplets (including cores) were generated simultaneously, as shown in Figure 7 (Appendix A). Because oil takes priority over water with regard to wetting of the PDMS surface, W/O droplets (water cores) were simply formed in the parallel flow. Then, the three-dimensional focusing flow allowed the oil to form droplets at the protrusion end.

The device using Module 1-B guaranteed a more uniform volume and number of cores in the W/O/W droplets. Because the liquids are independently injected in Module 1-A, whereas all the liquids are injected into the junction and the generation timing is passively synchronized in Module 1-B, the droplet morphology will be different depending on the module type.

In addition, the device using Module 1-A was more suitable for forming small and many cores with a thick oil shell, whereas that using Module 1-B was better for forming large cores with a thin shell layer. It was possible to form W/O/W droplets with a thin shell that the inner core could escape through.

In all cases, the total flow rates of the oil and inner water should be lower than the oil flow rate for only O/W droplet generation. Because the inner core reinforced the oil layer, deformation and drop-off of the oil layer (including the water core) were relatively difficult, and wetting of the oil on the channel surface occurred easily. The total volume of the W/O/W droplet was also comprehensively dependent on the inner core’s size and generation frequency, as well as on the flow rates of the oil and outer water. The droplet diameters generated under the suitable conditions were smaller than the hydraulic diameter of the downstream channel in the Module 3.

In these experiments, we showed the modulation and integration of the device elements, and that all directions (horizontal and vertical planes) were fully utilized for the functions of the elements. The functions of the device can be altered through modification of the independent modules and expanded by the addition of modules. Thus, this method would be widely applicable to multifunctional platforms that require complex functions and/or many processes in W/O/W droplet-based research.

### 4.3. Chemical Reaction in the W/O/W Droplets

Finally, the device with all three modules integrated was applied to the chemical synthesis of metal complexes in the W/O/W droplets. Mineral oil was used as the carrier flow in the first module, and methanol-based liquids were used as the sample phase flow. Then, the oil carrier formed shell-encapsulating methanol cores in the third module. The copper(II) acetate monohydrate and ligand were mixed in the alcohol-in-oil-in-water (A/O/W) droplets, and the copper complex was synthesized in the droplets. Synthesis of the copper complex in the micro-scale device was reported in our previous work [38], but the single-layer droplet’s reaction caused undesired sticking of crystals on the channel surface with long-term experiments and required a careful process for oil removal. Therefore, we applied this device to the complex synthesis for more efficient experiments.

As shown in Figure 8A, the encapsulation of the two reagents was successfully performed. However, mixing of the cores for the synthesis process was not clearly observable in the device because of the short channel length with the fast flow and the effect of the surfactant on the oil. Alternatively, the methanol cores in the droplets were mixed in a water pool outside of the device. The mixed cores formed crystals of the copper complex, and then the methanol core escaping from the oil shell during transfer was easily observed. Droplets immobilized between the cover glasses and crystals of the copper complex are shown in Figure 8B.

The inner cores had disappeared, but the crystals were trapped at the interface between the oil and outer water. The crystal morphology was almost the same as that of crystals synthesized in the droplets in our previous study [38], but the density and size of the crystals were different depending on the core volume and volume ratio of each reagent. Compared with the synthesis employing only alcohol-in-oil droplets, the undesired sticking of crystals on the channel surface, which causes wetting of alcohol droplet on the channel, was not observed in this A/O/W droplet-based synthesis method because the mixing and reaction had started in the double layer of the oil shell and water carrier. Furthermore, the small volume of the oil used as only the shell layer could reduce analytical error due to the oil.

To the best of our knowledge, this is the first report of the crystallization of a copper complex within a W/O/W droplet. The proposed structure allowed formation of the droplet morphology and the efficient conduction of experiments within the droplets without complicated chemical considerations. We believe that various experiments that were difficult to realize in conventional devices could be improved through the use of suitable reagents and materials, as well as consideration of the structural features of the device and the methods of droplet generation.

## 5. Conclusions

This paper presents a method for O/W and W/O/W droplet generation, as well as chemical synthesis in the droplets, through the full utilization of all dimensions in a three-dimensional fluidic device. The structures were fabricated by combining several functional modules, where the interlayer of the modules was not only used as a connection channel but also as a structure for the generation of highly wetting liquid droplets without the need for surface treatment and special additives. Whereas conventional droplet generators are only focused on the generation of the droplets, our proposed device allows for the simple installation of elements before and after the droplet-generating part. Moreover, the dimension of the device can be controlled more precisely and the generated droplet size was smaller compared with the results in glass capillary-based droplet generators.

In our precise device, droplets of highly wetting liquids were successfully formed under the fluidic conditions applied. On the basis of the structural feature, water-phase samples were independently formed and encapsulated in the oil shell. Furthermore, the number and volume of encapsulated inner cores were also controlled via the module design and fluidic conditions. Finally, the successful chemical reaction of the encapsulated samples within the droplets was verified.

This is the first report of the use of a vertical plane in a three-dimensional device that mainly has functions on the horizontal plane and is stacked in the vertical direction. In particular, droplet-based research requires complex functions, which can be achieved in this device. An additional advantage is that the oil and water droplets in the multilayered complex phases in the PDMS channel can be employed without any limitation of the materials used. The proposed structure allowed various types of oil droplet formation. The mineral, paraffin, and silicone oils are used for not only biological applications such as cell encapsulation and culture, but also medical, cosmetic, food, mechanical, and electrical industry as a functional material. Therefore, the proposed technology can be applied to the fields for the development of the emulsion, capsule, and particles, as well as a chemical platform evaluated in this research.

The last module was designed for inner cores’ mixing and observation, unfortunately, the mixing of the cores was not observed in the device. The short downstream channel allowed the mixing of inner cores at the outside of device, but appropriate channel designs or systems conventionally developed horizontal structures would be helpful for the full functions. This research focused mainly on the multiphase droplet generation. However, various conventional elements such as fluidic/electric circuits or sensors and actuators can be integrated in the single device by the method. Therefore, the proposed approach will improve and diversify the functions of wide research utilizing microfluidic devices, as well as microdroplet-based research.

## Figures and Tables

**Figure 1 micromachines-10-00613-f001:**
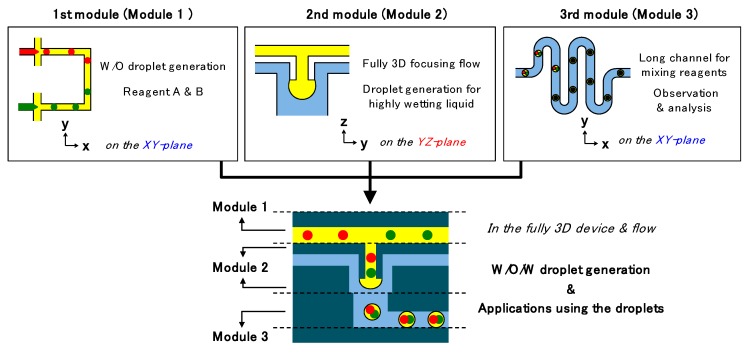
Schemes of the core-shell droplet-generating device formed by module integration; function and working plane of each module and fully integrated device.

**Figure 2 micromachines-10-00613-f002:**
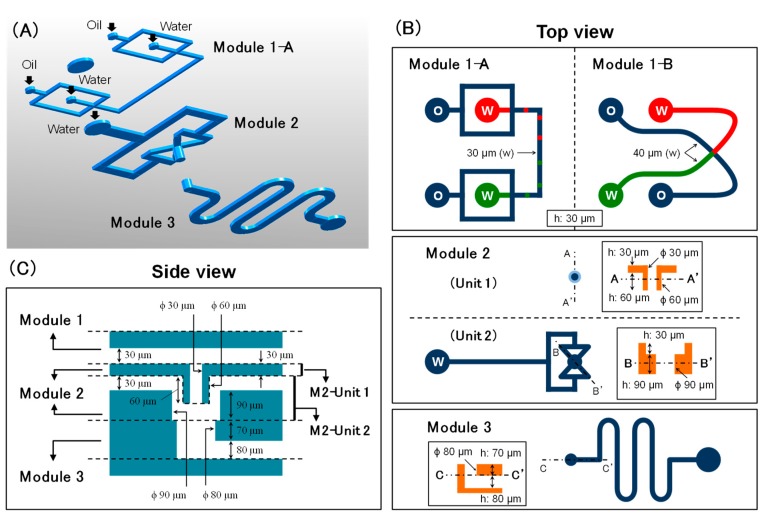
Schematic view and detailed dimensions of the three modules in the integrated device. (**A**) Bird’s-eye view of the full modules. (**B**) Top view of each module. (**C**) Side view of the integrated modules.

**Figure 3 micromachines-10-00613-f003:**
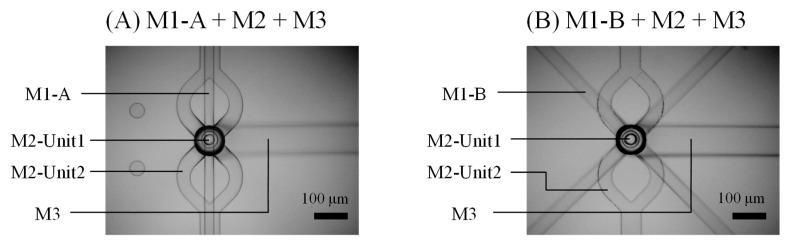
Top view of a junction area of the fully stacked three-dimensional device with Module 1-A (**A**) and Module 1-B (**B**) integrated.

**Figure 4 micromachines-10-00613-f004:**
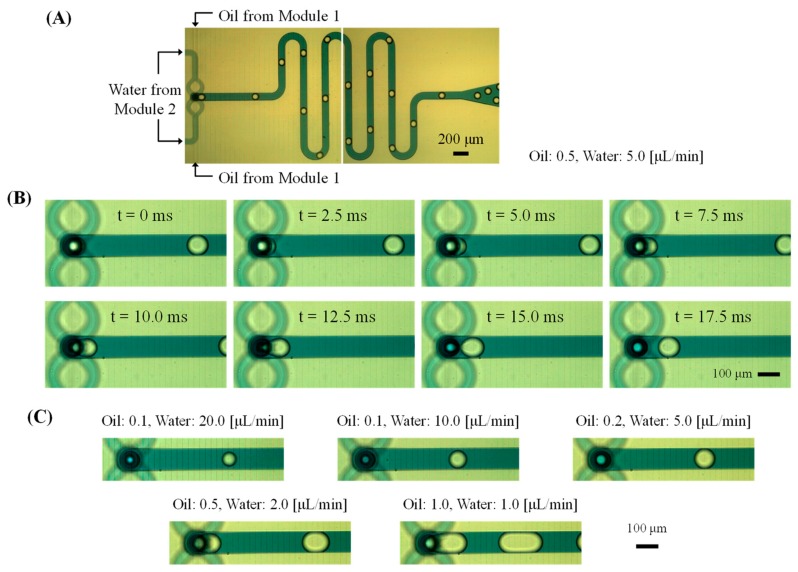
Captured images of mineral oil droplet generation in the fully integrated device. (**A**) Full view of the oil droplets in the Module 3. (**B**) Time-dependent behavior of the droplet generation at the junction of all modules. (**C**) Change of the droplet size via the fluidic conditions.

**Figure 5 micromachines-10-00613-f005:**
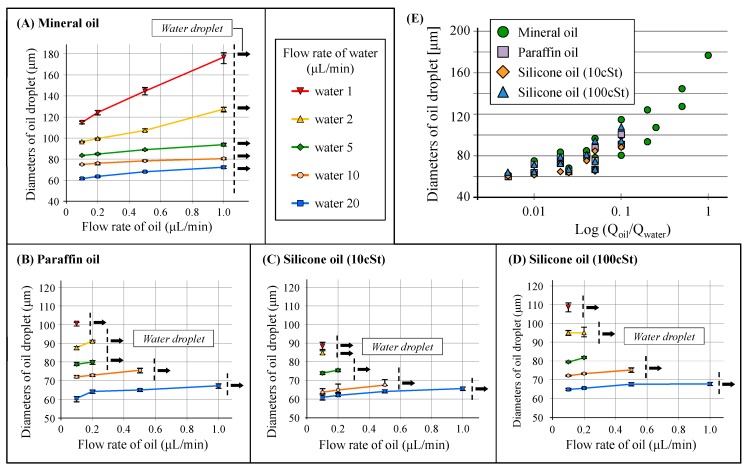
Diameters of the mineral oil (**A**), liquid paraffin oil (**B**), and silicone oil (**C**,**D**) droplets according to the flow conditions of the oil and water phases, and comparison of the droplet diameters generated under same flow conditions (**E**).

**Figure 6 micromachines-10-00613-f006:**
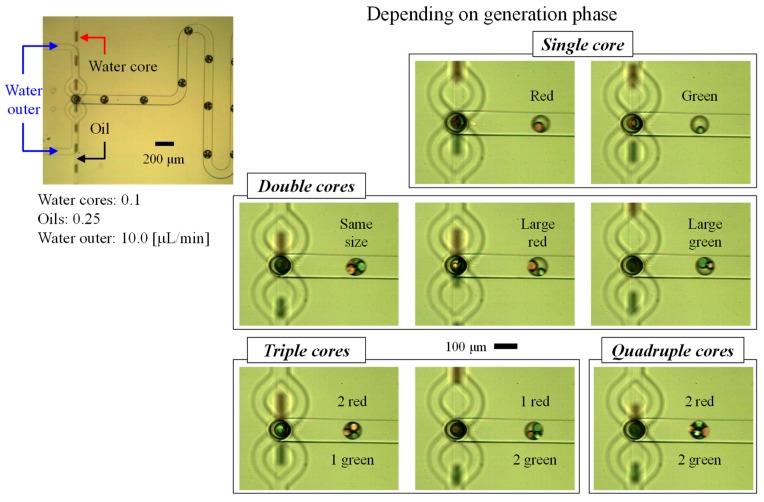
Captured images of the water-in-oil-in-water droplets generated from water-in-oil droplets and various core types in the droplets using the Type A device.

**Figure 7 micromachines-10-00613-f007:**
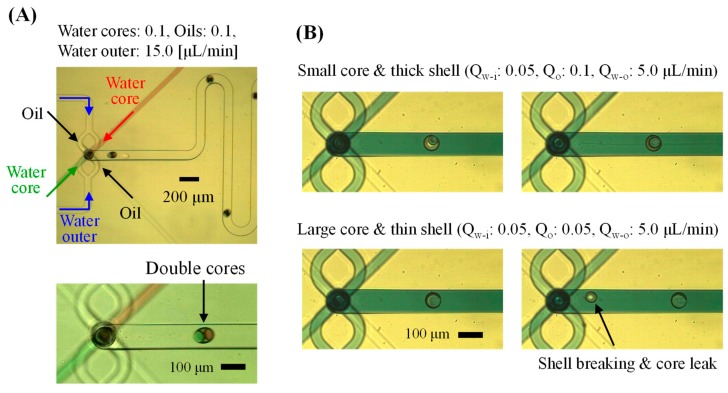
Water-in-oil-in-water (W/O/W) droplets generated using the Type B device. (**A**) Encapsulation of double cores in the W/O/W droplet. (**B**) Control of the water core volume and oil-shell thickness of the droplets.

**Figure 8 micromachines-10-00613-f008:**
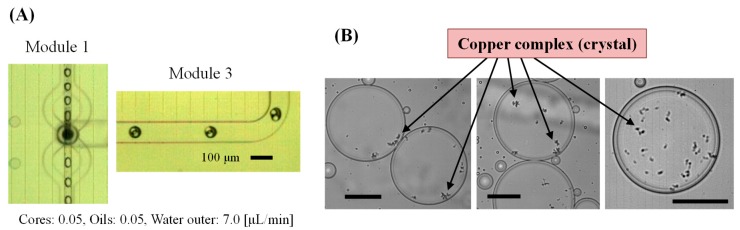
Copper complex synthesis in alcohol-in-oil-in-water droplets. Images of (**A**) droplet generation in the device and (**B**) synthesized crystals of copper complexes within the droplets (scale bars are 50 µm).

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
