# Peer review of "Integration of Horizontal and Vertical Microfluidic Modules for Core-Shell Droplet Generation and Chemical Application"

_micromachines, 2019, doi:10.3390/mi10090613_

Round 1

Reviewer 1 Report

Comments:

Yoon and co-workers reported the fabrication of a microfluidic setup able to produce water-in-oil-in-water droplets. The authors extended their methodology based on their previously published approach (Micromachines, 2018, 9, 468) for the formation of droplets with various numbers of cores and the synthesis of metal complexes. The authors approached this problem by fabricating three different modules for droplet generation, mixing and observation. In general, it is an interesting and novel approach for the formation of multiphase core-shell droplets with various applications. While interesting, this paper requires revisions before being suitable for publication in Micromachines. I suggest that the following points should be addressed:

The introduction needs improvement in terms of the language, context and references. The authors should get more specific and use appropriate references e.g lines 37-39, 68-70 and others. Please add appropriate references for the use of core-shell droplets for the synthesis of nanoparticles, microcapsules etc.  For Figure 5, do the authors have captured images of the droplets with bigger diameters d >100μm? Could they provide comments on the shape of droplets, spherical or elongated? That would provide also draw connections with the rest of the experiments (high flow rate ratios between outer and inner phases >10). Again, for Figure 5, I would also remove the composed lines for two experimental points, since there is no specific trend. In figure 6, single core – red: the authors claim that they have a single red droplet, however it seems that there is another tiny droplet next to it. Please address that or use another image. The authors present nice supplementary videos for the droplet generation. Could they also provide captured images or a video for the long-term stability of the droplet formation and travelling inside the microfluidic modules? That is particularly necessary for the reproducible synthesis of complex materials and droplet storage (relevant to their chemical synthesis). Figure 7B: It is hard to follow this figure from the text. Is there any flow rate-dependence? Also, what’s the connection between the two images of small core and thick shell (Figure 7B)? There are several issues regarding the microfluidic synthesis of metal complexes. Figure 8B illustrates the existence of very small W/O/W droplets (not discussed in the text). Are they formed during droplet generation, or after droplet storage? It is unclear what size, shape of copper complexes the authors are actually making. Is there any characterization? I suspect that small W/O/W droplets significantly affect the quality of the synthesis. Also, the fact that the synthesis doesn’t take place inside the device inherently limits the applicability of the device. Scale bars of the first two images for 8B need to be defined. The paper has a number of typographical errors, namely the word “Materials” is spelled “Materilas” (line 129). Please check and also spell check the entire paper to avoid poor English.

Author Response

Authors appreciate deeply for reviewers’ professional, kind, and detailed comments. We revised the manuscript and supplemented data as possible, based on the comments. Please consider the revision and let us know any additional comment. We hope this discussion will be helpful for potential readers of this research.

The response to the reviewer has attached as an MS word file. 

Reviewer 2 Report

The manuscript by Yoon and colleagues describes the design and fabrication of a three-dimensional microfluidic device (with horizontal and vertical planes) with individual modules that can altogether achieve one-step water/oil/water (W/O/W) or oil/water/oil (O/W/O) droplets. The authors demonstrate that they do not need channel treatment to form droplets and instead utilize the combination of horizontal and vertical features in the device coupled with precise control over the flow rates of each of the phases. The authors have adequately described about the device design and have also characterized it with respect to different oil media and different flow rates of the aqueous phase. And finally, an example application was also been presented. However, several major weakness exist in the current version of the manuscript which need to be addressed before it is ready for publication. A detailed list is included below:

Major Concerns

One of the main drawbacks of this paper is its presentation of figures and respective descriptions. There are many errors in the figures and legends (e.g., what are A and B in Fig.1?). Moreover, some of the figures are difficult to read due to small text (e.g., Figure 5). The authors need to revisit the figures and provide greater detail in the figure legends. Similarly, the description of the figures in the body of the manuscript is unclear at times which effects the overall presentation of the work. It is recommended that the authors perform a re-write to clarify the manuscript. The schematic of the whole device (comprising of the three modules) is missing in the manuscript. This reviewer would appreciate a schematic or cartoon in Figure 2 outlining the scheme of the whole device. In its current form, section 2 is unclear. Similarly, Section 2 (Principle and Design) could be improved as the current content lacks essential technical description. While some of the content is provided in the subsequent section (experimental) it is difficult to digest the information when it is presented out of order. This reviewer would recommending enhancing this section which should also include a more detailed and clearer explanation (e.g.., denoting which is oil and which is aqueous inlets/outlets? What are units 1 and 2 in module 2?) The authors have presented a case study in testing copper complex synthesis in A/O/W droplets; however the significance/impact of this system is not clearly presented. Can the authors explain why this case study was picked? Moreover, can the authors provide any quantitative analysis of the reaction (e.g., calculated rate constants). The data presented in Figure 5 lacks any metrics to confirm the statistical significance of the findings. It is recommended that additional analysis be performed to verify the statistical rigor of their findings. While the authors have done a good job in exploring different mineral oils and flowrates for both the phases, overall, this manuscript seems to present very early stage results that are mostly qualitative in nature. How broad are the applications of this device when there are several parameters involved? For example, how can two extremely viscous phases (e.g., gels and water), or any temperature/chemically sensitive component be handled on this platform? The authors should provide some context about the over utility of their system. Can the authors also provide some information on the overall throughput of their system in terms of droplet generation and droplet analysis? The authors mention that the force field within their device is different, but do not provide any calculations or simulations to support this claim.

Minor Concerns

Introduction needs referencing (lines 46-57). There are some typos which the author should be aware of (e.g., line 129). It is recommended that the author do some additional proofreading. The conclusions section could be enhanced to provide greater clarity regarding the conclusions and implications of this current study.

Author Response

(The authors gave the same response as above.)

Reviewer 3 Report

This manuscript deals with PDMS microchannel 3D structure that can generatemultiphases droplets and the characterization of the device by the flow conditions (i.e., flow rates). Authors built such 3D channel structure by modular method which is one of novel features of this study. Also using the device, authors demonstrated the chemical reaction of copper complex synthesis.

I enjoyed reading this manuscript. All necessary information are described well either in the body texts or in the figures. Figures and supplemental videos are self-explanatory and do not leave any vagueness behind.

Minor suggestions & comments

1) In Fig.1, 3rd module is illustrated as the module for mixing of reagents A and B. However, none of the result figures (6, 7 , or 8) show the mixing stage of droplet. Perhaps, Module 3 photo of Fig. 8 may be a good place to show mixing? 

2) In Fig. 5, some droplets had the larger diameter than the width of Module 3 channel, from which I guessed the droplet shape was not circular. Since oil wet PDMS surface much better than water, the stability of droplet was questioned (i.e., oil droplet wet the channel wall and outer water broke the oil droplet?)

3) In the discussion of the results shown in Fig. 5, can the capillary number of oil which might be providing droplet generation criteria be added?

Author Response

(The authors gave the same response as above.)

Round 2

Reviewer 1 Report

I appreciate the authors' detailed responses and changes in the manuscript. They addressed most of my comments with changes in the main text and additional figures. I suggest that the manuscript should be accepted at its current form. 

Author Response

The authors appreciate deeply the reviewer’s professional and kind comments again.

Reviewer 2 Report

This reviewer thanks the authors for their efforts in the revision to address the concerns. While this version is improved, this reviewer still has a few minor concerns that should be addressed. These are detailed below:

In reference to the comments 2 and 3 from the initial review, this reviewer still finds section 2 to be unclear with the presented description. It is understandable that the novelty of this work is just module 2, but the authors could explain a bit about modules 1 and 3 instead of just mentioning these as conventionally developed structures (also a reference is missing here in line 93 if the authors are citing their previous work). The authors could start with explaining module 1 through 3 sequentially to achieve a better flow. A general comment about all the figures: It would be easier for the readers to follow the figures if the authors labelled each part individually (with A,B,C/i, ii, iii). Also, the legends of the figures could have a few lines of description rather than just putting forth a statement. The authors’ response for comment 4 as to why they have picked this case study does not fully address this reviewers’ concerns. Could the authors please provide a more comprehensive response, especially providing insight in the manuscript? The authors could start by explaining the significance of this application, followed by what was done in the past and what has been done/analyzed in this work. In reference to comment 6, the authors are explaining the properties of different oils, but are not explaining about the potential applications of this device. A fundamental technique like this will always have to be able to find a broad range of applications. The authors mention that the potential future work will be able to find all these applications which leads to questioning the purpose of this current work. This reviewer would recommend that the authors include the throughput information of the device (e.g., an average number for the best-case scenario), since this would be a very basic question when trying to implement this device design elsewhere.

Author Response

The authors appreciate deeply the reviewer’s professional and kind comments again. The response to the comments was attached as an MS word file. We revised the manuscript and hope the revision is helpful for potential readers of this research.

Reviewer 3 Report

Authors responded all the questions by this reviewer and the revised manuscript looks satisfactory to be published as the current form.

Author Response

(The authors gave the same response as above.)
